# Learning to Reconstruct Missing Data from Spatiotemporal Graphs with Sparse Observations

**Ivan Marisca***
The Swiss AI Lab IDSIA
Università della Svizzera italiana
ivan.marisca@usi.ch

**Andrea Cini***
The Swiss AI Lab IDSIA
Università della Svizzera italiana
andrea.cini@usi.ch

**Cesare Alippi**
The Swiss AI Lab IDSIA
Università della Svizzera italiana
Politecnico di Milano
cesare.alippi@usi.ch

## Abstract

Modeling multivariate time series as temporal signals over a (possibly dynamic) graph is an effective representational framework that allows for developing models for time series analysis. Spatiotemporal graphs are often highly sparse, with time series characterized by multiple, concurrent, and even long sequences of missing data, e.g., due to the unreliable underlying sensor network. In this context, autoregressive models can be brittle and exhibit unstable learning dynamics. The objective of this paper is to tackle the problem of learning effective models to reconstruct, i.e., *impute*, missing data points by conditioning the reconstruction only on the available observations. In particular, we propose a novel class of attention-based architectures that, given a set of highly sparse discrete observations, learn a representation for points in time and space by exploiting a spatiotemporal propagation architecture aligned with the imputation task. Representations are learned end-to-end to reconstruct observations w.r.t. the corresponding sensor and its neighboring nodes. Compared to the state of the art, our model handles sparse data without propagating prediction errors or requiring a bidirectional model to encode forward and backward time dependencies. Empirical results on representative benchmarks show the effectiveness of the proposed method.

## 1 Introduction

Exploiting structure – both temporal and spatial – is arguably the key ingredient for the success of modern deep-learning architectures and models. This is the case with spatiotemporal graph neural networks (STGNNs) [1–3], which learn to process multivariate time series while taking into account underlying space and time dependencies by encoding structural spatiotemporal inductive biases in their architectures. However, even when spatiotemporal relationships are present, available data are almost always incomplete and irregularly sampled, both spatially and temporally. This is definitely true for data coming from real sensor networks (SNs), where missing time series observations are usually imputed with simple interpolation strategies before proceeding with the downstream task. More advanced methods deal with missing data by autoregressively replacing missing observations with predicted ones, eventually using bidirectional architectures [4, 5] to exploit both forward and backward temporal dependencies. To account also for spatial dependencies, Cini et al. [6] introduced a method, named GRIN, combining a bidirectional autoregressive architecture with message passing

---

*Equal contribution.

I. Marisca et al., Learning to Reconstruct Missing Data from Spatiotemporal Graphs with Sparse Observations (Extended Abstract). Presented at the First Learning on Graphs Conference (LoG 2022), Virtual Event, December 9–12, 2022.

neural networks [7–10]. Despite being the state of the art in spatiotemporal imputation, GRIN can suffer from the error propagation typical of autoregressive models [11, 12]. In fact, we argue that the propagation of imputed (biased) values through space and time combined with noisy observations might exacerbate error accumulation in highly sparse data and drive the hidden state of GRIN-like models to drift away.

In this paper, we aim at tackling this problem by designing an architecture based on a novel attention mechanism that takes spatiotemporal sparsity into account while learning representations and imputing missing values. Compared with the alternatives discussed so far, our method exploits a novel spatiotemporal propagation process to learn a predictive representation for each missing observation by relying only on observed values propagated through the spatiotemporal structure. This approach achieves the twofold objective of avoiding propagating biased representation – typical in the autoregressive framework – and reconstructing observations at arbitrary nodes in the sensor network. In summary, our main contributions are:

1. the introduction of a sparse spatiotemporal attention mechanism to learn, from sparse data, representations localized in time and space;

2. the design of a novel STGNN based on the aforementioned spatiotemporal attention mechanism and equipped with inductive biases that make the model tailored for the multivariate time series imputation task;

3. an empirical assessment of the proposed method showing how it overcomes the limits of existing approaches, particularly in settings with highly sparse data.

## 2 Problem formulation and related works

We denote by $\boldsymbol{X}_t \in \mathbb{R}^{N \times d}$ the matrix collecting the $d$-dimensional measurements of $N$ sensors (or measurement stations) in a SN at time step $t$, with $\boldsymbol{X}_{t:t+T}$ being the sequence of $T$ measurements collected in the time interval $[t, t+T]$. We model functional relationships among the sensors as graph edges, represented by the weighted adjacency matrix $\boldsymbol{A} \in \mathbb{R}^{N \times N}$, in which each nonzero entry $a^{i,j}$ indicates the weight of the edge going from the $i$-th node to the $j$-th. We assume to have available sensor-level covariates $\boldsymbol{Q}_t \in \mathbb{R}^{N \times d_q}$ that act as spatiotemporal coordinates to localize a point in time and space (e.g., date/time features and geographic location). To account for data availability, we use a binary mask $\boldsymbol{m}_t \in \{0, 1\}^N$ whose $i$-th element $m_t^i$ is 1 if the measurements associated with the $i$-th sensor are valid at time step $t$. Conversely, if $m_t^i = 0$, we consider the measurements $\boldsymbol{x}_t^i$ to be completely missing, with the exogenous variables $\boldsymbol{q}_t^i$ being instead available. Finally, we model the multivariate, structured time series as a discrete sequence of graphs, where each graph is a tuple $\mathcal{G}_t = \langle \boldsymbol{X}_t, \boldsymbol{Q}_t, \boldsymbol{m}_t, \boldsymbol{A} \rangle$. Denoting by $\widetilde{\boldsymbol{X}}_{t:t+T}$ the unknown corresponding complete sequence, the goal of multivariate time series imputation (MTSI) is to find an estimate $\widehat{\boldsymbol{X}}_{t:t+T}$ minimizing the reconstruction error over the missing data points. Notice that, since $\widetilde{\boldsymbol{X}}_{t:t+T}$ is not available, one should find a surrogate objective or simulate the presence of missing data, for which the reconstruction error can be computed.

**Related works.** Multivariate time series imputation is a core task in time series analysis and deep learning methods are commonly used in this regard. In particular, deep autoregressive models based on recurrent neural networkss (RNNs) are currently among the most widely adopted methods [4, 5, 13, 14]. Several approaches in the literature, then, rely on generative adversarial neural networks [15] to generate imputed subsequences by matching the underlying data distribution [14, 16, 17]. Recently, several attention-based imputation techniques have also been proposed [18–20]. More related to our work, GRIN [6] uses a bidirectional graph RNN with a message passing spatial decoder, to impute time series based on spatiotemporal dependencies. The attention mechanism [21, 22] has been exploited in several contexts within the graph deep learning literature [23–26]. In particular, TraverseNet [27] is specially related to our work, since it relies on spatiotemporal autoregressive attention to compute messages exchanged between nodes.

## 3 Methodology

The autoregressive approach to reconstruction consists in directly modeling distributions $p\left(\boldsymbol{x}_t^i \mid \boldsymbol{X}_{<t}\right)$, with $\boldsymbol{X}_{<t}$ being the sequence of observations prior to time step $t$, and using one-step-ahead forecasting

as a surrogate objective to learn how to recover missing observations. To also consider $\boldsymbol{X}_{>t}$, i.e., data subsequent to the target time step, it is common to use a bidirectional architecture which models also $p\big(\boldsymbol{x}_t^i \,|\, \boldsymbol{X}_{>t}\big)$ [5, 28]. Moreover, a third component $p\big(\boldsymbol{x}_t^i \,|\, \{\boldsymbol{x}_t^{j\neq i}\}\big)$ must be introduced to account for spatial information at each step. Architectures like GRIN, follow exactly this scheme with different components dedicated to model each of these three aspects. While being effective in practice, these approaches can have multiple drawbacks. Besides the computational overhead of having three separate components and the compounding error in the autoregressive models [11, 12], they can fall short in capturing global context, as the processing of the structural information is decomposed. Furthermore, merging the information coming from the different modules is also problematic, yielding to further compounding of errors. Finally, in the case of highly sparse observations, the spatial processing should be dealt with special care as propagating information through partially observed spatiotemporal graphs adds another layer of complexity.

Our approach, named *Spatiotemporal Point Inference Network* (SPIN), is a graph attention network for MTSI, designed to learn representations of discrete points associated with nodes of a sequence of spatiotemporal graphs. We denote as *observed set* $\mathcal{X}_{t:t+T} = \big\{\big\langle \boldsymbol{x}_\tau^i, \boldsymbol{q}_\tau^i \big\rangle \,|\, m_\tau^i = 1 \wedge \tau \in [t, t+T)\big\}$ the set of all observations, paired with their spatiotempotal coordinates. Conversely, we name *target set* $\mathcal{Y}_{t:t+T} = \big\{\boldsymbol{q}_\tau^i \,|\, m_\tau^i = 0 \wedge \tau \in [t, t+T)\big\}$ the complement set collecting the coordinates of the discrete spatiotemporal points for which we want to reconstruct an observation. Then, for all discrete points $\boldsymbol{q}_\tau^i \in \mathcal{Y}_{t:t+T}$, SPIN is trained to learn a model

$$f_\theta(\boldsymbol{q}_\tau^i \,|\, \mathcal{X}_{t:t+T}, \boldsymbol{A}) \approx \mathbb{E}\big[p\big(\boldsymbol{x}_\tau^i \,|\, \boldsymbol{q}_\tau^i, \mathcal{X}_{t:t+T}, \boldsymbol{A}\big)\big]. \tag{1}$$

To this end, SPIN learns a parameterized propagation process where each representation, corresponding to a node and time step, is updated by aggregating information from all the available observations acquired at neighboring nodes, weighted by input-dependent attention scores. The core component of SPIN is a novel *sparse spatiotemporal attention* layer (Figure 1) used to propagate information at the level of single observations. Indeed, leveraging on the attention mechanism, we learn representations for each $i$-th node at each $\tau$-th time step by simultaneously aggregating information from (1) the observed set of $i$-th node $\mathcal{X}_{t:t+T}^i$; (2) the observed set $\mathcal{X}_{t:t+T}^j$ of each $j \in \mathcal{N}(i)$, i.e., the set of neighbors of the $i$-th node.

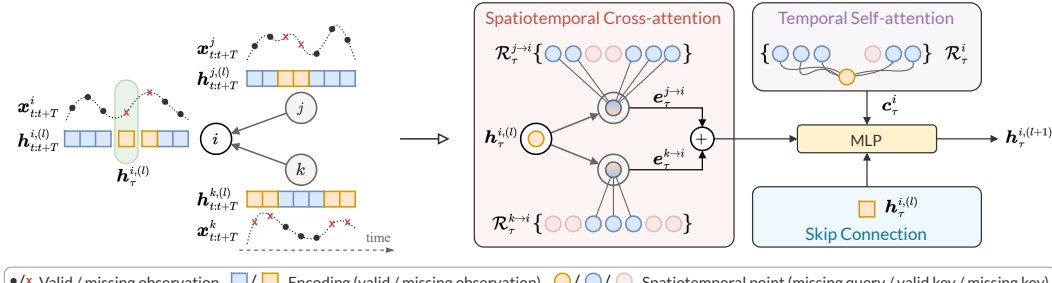

Figure 1: Example of the sparse spatiotemporal attention layer acting for updating $\boldsymbol{h}_\tau^{i,(l)}$, by simultaneously performing inter-node spatiotemporal cross-attention and intra-node temporal self-attention.

Let $\boldsymbol{h}_\tau^{i,(l)} \in \mathbb{R}^{d_h}$ be the learned representation for the $i$-th node and time step $\tau$ at the $l$-th layer. The encoding is initialized as $\mathrm{MLP}\big(\boldsymbol{x}_\tau^i, \boldsymbol{q}_\tau^i\big)$ if observation $\boldsymbol{x}_\tau^i$ is valid, or $\mathrm{MLP}\big(\boldsymbol{q}_\tau^i\big)$ otherwise, where MLP is a multi-layer perceptron. The next steps involve computations of spatiotemporal messages, i.e., representations computed to propagate information from one discrete space-time point to another. We indicate the propagation along the temporal dimension from time step $s$ to time step $\tau$ as subscripts $s \to \tau$. Similarly, superscripts $j \to i$ indicate messages sent from the $j$-th node to the $i$-th. To avoid overloading the notation, we omit the layer superscript in the following. The message $\boldsymbol{r}_{s\to\tau}^{j\to i} \in \mathbb{R}^{d_h}$ from the $j$-th node at time step $s$ to the $i$-th node at time step $\tau$ is computed with an MLP taking as input both source and target representations (Eq. 2). To account for spatial information, this mechanism is used to perform an *inter-node* temporal cross-attention, computing a message to $\boldsymbol{h}_\tau^i$ using encodings in $\boldsymbol{h}_{t:t+T}^j$ associated with a valid observation for every neighbor $j \in \mathcal{N}(i)$ (Eq. 3).

$$\boldsymbol{r}_{s\to\tau}^{j\to i} = \mathrm{MLP}\big(\boldsymbol{h}_s^j, \boldsymbol{h}_\tau^i\big) \qquad (2) \qquad \mathcal{R}_\tau^{j\to i} = \{\boldsymbol{r}_{s\to\tau}^{j\to i} \,|\, \big\langle \boldsymbol{x}_s^j, \boldsymbol{q}_s^j \big\rangle \in \mathcal{X}_{t:t+T}\} \tag{3}$$

Messages in $\mathcal{R}_\tau^{j\to i}$ are then weighted by *message scores* $\alpha_{s\to\tau}^{j\to i}$, computed by a linear projection of

**Table 1:** Performance (MAE) with increasing data sparsity (average over $5$ evaluation masks).

| | METR-LA | | | PEMS-BAY | | | AQI | | |
|---|---|---|---|---|---|---|---|---|---|
| | 5 % | 10 % | 15 % | 5 % | 10 % | 15 % | 5 % | 10 % | 15 % |
| BRITS | $5.87 \pm 0.03$ | $7.26 \pm 0.06$ | $8.29 \pm 0.07$ | $4.14 \pm 0.05$ | $5.41 \pm 0.08$ | $5.84 \pm 0.04$ | $24.09 \pm 0.30$ | $31.90 \pm 0.26$ | $37.62 \pm 0.42$ |
| SAITS | $4.73 \pm 0.07$ | $6.66 \pm 0.05$ | $7.27 \pm 0.03$ | $3.88 \pm 0.09$ | $7.62 \pm 0.21$ | $8.01 \pm 0.11$ | $20.78 \pm 0.30$ | $30.16 \pm 0.39$ | $36.34 \pm 0.33$ |
| Transformer | $6.03 \pm 0.04$ | $7.19 \pm 0.05$ | $8.06 \pm 0.05$ | $3.69 \pm 0.06$ | $5.09 \pm 0.05$ | $6.02 \pm 0.04$ | $29.21 \pm 0.33$ | $33.62 \pm 0.16$ | $37.31 \pm 0.14$ |
| GRIN | $3.05 \pm 0.02$ | $4.52 \pm 0.05$ | $5.82 \pm 0.06$ | $2.26 \pm 0.03$ | $3.45 \pm 0.06$ | $4.35 \pm 0.04$ | $15.62 \pm 0.24$ | $22.08 \pm 0.39$ | $29.03 \pm 0.42$ |
| **SPIN** | $2.71 \pm 0.02$ | $3.32 \pm 0.02$ | $3.87 \pm 0.05$ | $\mathbf{1.78 \pm 0.03}$ | $\mathbf{2.15 \pm 0.03}$ | $\mathbf{2.41 \pm 0.02}$ | $\mathbf{14.29 \pm 0.24}$ | $\mathbf{18.71 \pm 0.34}$ | $\mathbf{24.34 \pm 0.46}$ |
| **SPIN-H** | $\mathbf{2.64 \pm 0.02}$ | $\mathbf{3.17 \pm 0.02}$ | $\mathbf{3.61 \pm 0.04}$ | $\mathbf{1.75 \pm 0.04}$ | $2.16 \pm 0.03$ | $2.48 \pm 0.02$ | $14.55 \pm 0.26$ | $19.37 \pm 0.36$ | $25.38 \pm 0.37$ |

the messages in $\mathcal{R}_\tau^{j \to i}$ followed by a softmax layer, and aggregated to obtain an edge-level context vector $\boldsymbol{e}_\tau^{j \to i}$, encoding the observed sequence at each $j$-th node w.r.t. the $i$-th node and time step $\tau$. Analogously, to account for the observed sequence of the $i$-th node itself, we exploit an *intra-node* temporal self-attention mechanism to compute messages from the encodings $\boldsymbol{h}_{t:t+T}^i$ corresponding to valid observations, aggregated (weighted by message scores) to obtain a temporal context vector $\boldsymbol{c}_\tau^i$. Then, target representation $\boldsymbol{h}_\tau^{i,(l)}$ is updated with a final aggregation step (Eq. 4), and imputations for all spatiotemporal points in $\mathcal{Y}_{t:t+T}$ are obtained – after $L$ layers – with a nonlinear readout (Eq. 5).

$$\boldsymbol{h}_\tau^{i,(l+1)} = \text{MLP}\Big( \boldsymbol{h}_\tau^{i,(l)}, \ \boldsymbol{c}_\tau^{i,(l)}, \ \sum_{j \in \mathcal{N}(i)} \boldsymbol{e}_\tau^{j \to i,(l)} \Big) \tag{4}$$

$$\widehat{\mathcal{Y}}_{t:t+T} = \{ \hat{\boldsymbol{x}}_\tau^i = \text{MLP}\big( \boldsymbol{h}_\tau^{i,(L)} \big) \,|\, \boldsymbol{q}_\tau^i \in \mathcal{Y}_{t:t+T} \} \tag{5}$$

**Hierarchical attention.** Roughly speaking, the proposed spatiotemporal attention mechanism can be viewed as performing attention over the spatiotemporal graph $\mathcal{S}$, obtained by considering the product graph between space and time dimensions. Performing graph attention on the surrogate graph $\mathcal{S}$ has time and memory complexities that scale with $O((N + E)T^2)$, with $N, E$ being the largest number of nodes and edges, respectively, among graphs in $\mathcal{G}_{t:t+T}$. To reduce this computational burden – which undermines the application of the proposed method to large graphs and long time horizons – we propose to rewire the attention mechanism to be hierarchical [29]. We do this by adding $K$ dummy nodes that act as hubs for propagating information. In this way, we can reduce the spatiotemporal attention complexity to $O((N + E)KT)$, with $K \ll T$, at the cost of introducing an information bottleneck. We refer to Appendix B for a detailed explanation of this mechanism.

## 4 Empirical evaluation

In this section, we evaluate our method on three real-world datasets and compare the performance against state-of-the-art methods and standard approaches for MTSI. In following experiment, we consider both **SPIN** and the hierarchical version **SPIN-H**. The figure of merit used is the *mean absolute error* (MAE), averaged across imputation windows. We consider only the *out-of-sample* scenario similarly to previous works [6], in which every parametric model is trained and tested on disjoint sets. We consider three openly available datasets coming from real-world SNs. The first two, namely **PEMS-BAY** and **METR-LA** [2], record traffic measurements and are both widely used benchmarks in spatiotemporal forecasting literature. We use the same setup of [6] to inject missing data with *Point missing* policy, in which we randomly drop $25\%$ of the available data. As a third dataset, we consider **AQI** [30, 31], which collects hourly measurements of air pollutants from $437$ air quality monitoring stations in China. We consider also a smaller version of this dataset (**AQI-36**) with only the 36 sensors in the city of Beijing. We use the same missing data distribution used in [6, 31]. In all settings, all the valid observations masked out are used as targets for evaluation. We obtain an adjacency matrix from the pairwise distance of sensors following previous works [2, 3, 6]. We compare our methods against (1) GRIN [6], a graph-based bidirectional RNN for MTSI with state-of-the-art performance; (2) a spatiotemporal Transformer, where we alternate temporal and spatial Transformer encoder layers from [21] and replace missing values with a [MASK] token (as in [32]); (3) SAITS [18], a recent attention-based architecture; (4) BRITS [5], which leverages on a bidirectional RNN. We assess how performance changes as the percentage of missing values increases. In practice, we change the missing data distribution at test time, simulating the case in which, at each time step, every sensor has a constant probability $\bar{p}$ of going offline for a random number $S \sim \mathcal{U}(12, 36)$ of future (consecutive) time steps. Table 1 shows results for all datasets with $\bar{p} = 5\%$, $\bar{p} = 10\%$, and

$\bar{p} = 15\%$. Note that, depending on the dataset, the portion of valid observations in each of these cases amounts to $\approx$ 25-30%, $\approx$ 8-10%, and $\approx$ 3-4%, respectively. SPIN models, differently from the baselines, can handle all the considered scenarios. In particular, improvements in performance w.r.t. the best performing baseline (GRIN) are more evident as the number of available observations decreases. Indeed, the sparse spatiotemporal attention mechanism of SPIN is not autoregressive and allows an unbounded memory capacity. Note also that SPIN-H performs on par (and in some cases better) with SPIN, making it a valid lightweight alternative to SPIN. In Appendix A, we show that SPIN-based models perform on par or better than state-of-the-art methods on standard benchmarks. The code to reproduce the experiments of the paper is available online[2].

## 5 Conclusions

We introduced a graph-based attention network, named SPIN, to reconstruct missing observations in sparse spatiotemporal time series. We showed how the time and space complexities of the approach can be drastically reduced by considering a novel hierarchical attention mechanism. Empirical analysis shows that the proposed method widely outperforms state-of-the-art methods for imputation in highly sparse settings. Future works could investigate the application of SPIN in other time series analysis tasks (e.g., forecasting), as well as in settings with an underlying dynamic graph topology.

## Author Contributions

The authors' contributions to this work are specified as follows, adopting the Contributor Roles Taxonomy (CRediT):

- Ivan Marisca: Conceptualization, Data curation, Formal Analysis, Investigation, Methodology, Software, Validation, Visualization, Writing – original draft, Writing – review & editing.
- Andrea Cini: Conceptualization, Data curation, Formal Analysis, Investigation, Methodology, Software, Validation, Visualization, Writing – original draft, Writing – review & editing.
- Cesare Alippi: Conceptualization, Formal Analysis, Funding acquisition, Methodology, Project administration, Resources, Supervision, Writing – original draft, Writing – review & editing.

## Acknowledgements

This work was supported by the Swiss National Science Foundation project FNS 204061: *Higher-Order Relations and Dynamics in Graph Neural Networks*. The authors wish to thank the Institute of Computational Science at USI for granting access to computational resources.

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

# Appendix

## A  Performance on standard benchmarks

Table 1, in the main paper, shows the reconstruction error of the different methods as the number of valid observations in input sequences decreases. To assess the performance of our model in standard settings (denser observations), we test all the methods on the original datasets introduced in section 4. For the traffic datasets, we also consider a different evaluation mask, where data are removed according to the *Block missing* policy [6], in which we randomly mask out $5\%$ of the available data and, in addition, we simulate failures of $S \sim \mathcal{U}(12, 48)$ consecutive steps with $0.15\%$ probability. For this experiment, we consider also additional baselines: (1) node-level sequence mean (MEAN); (2) neighbors mean (KNN); (3) Matrix Factorization (MF); (4) MICE [33]; (5) VAR, a vector autoregressive one-step-ahead predictor; (6) rGAIN, an adversarial approach which shares similarities with GAIN [16] and SSGAN [14]. Table 2 shows the results in terms of MAE. Whenever possible, we use results from [6].

**Table 2:** Performance (in terms of MAE) averaged over multiple independent runs.

| | Block missing | | Point missing | | Simulated failures | |
|---|---|---|---|---|---|---|
| | PEMS-BAY | METR-LA | PEMS-BAY | METR-LA | AQI-36 | AQI |
| Mean | $5.46 \pm 0.00$ | $7.48 \pm 0.00$ | $5.42 \pm 0.00$ | $7.56 \pm 0.00$ | $53.48 \pm 0.00$ | $39.60 \pm 0.00$ |
| KNN | $4.30 \pm 0.00$ | $7.79 \pm 0.00$ | $4.30 \pm 0.00$ | $7.88 \pm 0.00$ | $30.21 \pm 0.00$ | $34.10 \pm 0.00$ |
| MF | $3.28 \pm 0.01$ | $5.46 \pm 0.02$ | $3.29 \pm 0.01$ | $5.56 \pm 0.03$ | $30.54 \pm 0.26$ | $26.74 \pm 0.24$ |
| MICE | $2.94 \pm 0.02$ | $4.22 \pm 0.05$ | $3.09 \pm 0.02$ | $4.42 \pm 0.07$ | $30.37 \pm 0.09$ | $26.98 \pm 0.10$ |
| VAR | $2.09 \pm 0.10$ | $3.11 \pm 0.08$ | $1.30 \pm 0.00$ | $2.69 \pm 0.00$ | $15.64 \pm 0.08$ | $22.95 \pm 0.30$ |
| rGAIN | $2.18 \pm 0.01$ | $2.90 \pm 0.01$ | $1.88 \pm 0.02$ | $2.83 \pm 0.01$ | $15.37 \pm 0.26$ | $21.78 \pm 0.50$ |
| BRITS | $1.70 \pm 0.01$ | $2.34 \pm 0.01$ | $1.47 \pm 0.00$ | $2.34 \pm 0.00$ | $14.50 \pm 0.35$ | $20.21 \pm 0.22$ |
| SAITS | $1.56 \pm 0.01$ | $2.30 \pm 0.01$ | $1.40 \pm 0.03$ | $2.26 \pm 0.00$ | $18.16 \pm 0.42$ | $21.33 \pm 0.15$ |
| Transformer | $1.70 \pm 0.02$ | $3.54 \pm 0.00$ | $0.74 \pm 0.00$ | $2.16 \pm 0.00$ | $11.98 \pm 0.53$ | $18.11 \pm 0.25$ |
| GRIN | $1.14 \pm 0.01$ | $2.03 \pm 0.00$ | $\mathbf{0.67} \pm \mathbf{0.00}$ | $\mathbf{1.91} \pm \mathbf{0.00}$ | $12.08 \pm 0.47$ | $14.73 \pm 0.15$ |
| **SPIN** | $\mathbf{1.06} \pm \mathbf{0.01}$ | $\mathbf{1.97} \pm \mathbf{0.01}$ | $0.71 \pm 0.01$ | $\mathbf{1.90} \pm \mathbf{0.01}$ | $11.77 \pm \mathbf{0.74}$ | $\mathbf{14.00} \pm \mathbf{0.13}$ |
| **SPIN-H** | $\mathbf{1.06} \pm \mathbf{0.01}$ | $2.05 \pm 0.03$ | $0.74 \pm 0.02$ | $1.96 \pm 0.04$ | $\mathbf{11.08} \pm \mathbf{0.06}$ | $14.39 \pm 0.03$ |

Both SPIN methods outperform the baselines in almost all scenarios. As expected, improvements are more evident when entire blocks of data are missing, as in AQI datasets and block missing settings. With respect to the spatiotemporal Transformer, SPIN performs better in all settings except for AQI-36, which can be attributed to the ineffectiveness of spatial attention alone in determining the dependencies among the different spatial locations.

## B  Hierarchical attention

In section 3 we introduced a hierarchical attention mechanism to reduce the computational complexity of the spatiotemporal attention mechanism in SPIN. In particular, such a mechanism acts by adding $K$ hub nodes that selectively propagate information through the graph. Let $\boldsymbol{Z}^i \in \mathbb{R}^{K \times d_z}$ be the hub nodes' representations for central node $i$, and then, for hub $k$ proceed as follows.

1. Update $\boldsymbol{z}_k^i$ by querying $\{\boldsymbol{h}_\tau^i \,|\, \langle \boldsymbol{x}_\tau^i, \boldsymbol{q}_\tau^i \rangle \in \mathcal{X}_{t:t+T}\}$, i.e., node encodings associated with valid observations, obtaining $\tilde{\boldsymbol{z}}_k^i$;

2. Update node encoding $\boldsymbol{h}_\tau^i$ by querying updated $\widetilde{\boldsymbol{Z}}^i$ and $\widetilde{\boldsymbol{Z}}^j$ of every $j$-th neighbor in $\mathcal{N}(i)$.

The spatiotemporal attention is effectively split into two phases. At first, we update each hub node representation similarly as done for the edge-level and temporal context vectors:

$$\boldsymbol{r}_{\tau,k}^i = \text{MLP}\big(\boldsymbol{h}_\tau^i, \boldsymbol{z}_k^i\big) \qquad (6)$$

$$\mathcal{R}_k^i = \{\boldsymbol{r}_{\tau,k}^i \,|\, \langle \boldsymbol{x}_\tau^i, \boldsymbol{q}_\tau^i \rangle \in \mathcal{X}_{t:t+T}\} \qquad (7)$$

$$\boldsymbol{c}_k^i = \sum_{\tau : \boldsymbol{r}_{\tau,k}^i \in \mathcal{R}_k^i} \alpha_{\tau,k}^i \cdot \boldsymbol{r}_{\tau,k}^i \qquad (8)$$

$$\tilde{\boldsymbol{z}}_k^i = \text{MLP}\big(\boldsymbol{z}_k^i, \boldsymbol{c}_k^i\big) \qquad (9)$$

**Figure 2:** The architecture of SPIN. At first, we encode observations $\boldsymbol{X}_{t:t+T}$ and spatiotemporal coordinates $\boldsymbol{Q}_{t:t+T}$, obtaining initial representations $\boldsymbol{H}_{t:t+T}^{(0)}$. The representations are updated by a stack of $L$ sparse spatiotemporal attention blocks. Final imputations are obtained from $\boldsymbol{H}_{t:t+T}^{(L)}$ with a nonlinear readout.

Then, we obtain context vectors from the updated hub representations as:

$$\boldsymbol{r}_{k,\tau}^{i} = \text{MLP}\big(\tilde{\boldsymbol{z}}_{k}^{i}, \boldsymbol{h}_{\tau}^{i}\big) \qquad (10) \qquad\qquad \boldsymbol{r}_{k,\tau}^{j \to i} = \text{MLP}\big(\tilde{\boldsymbol{z}}_{k}^{j}, \boldsymbol{h}_{\tau}^{i}\big) \qquad (12)$$

$$\boldsymbol{c}_{\tau}^{i} = \sum_{k} \alpha_{k,\tau}^{i} \cdot \boldsymbol{r}_{k,\tau}^{i} \qquad (11) \qquad\qquad \boldsymbol{e}_{\tau}^{j \to i} = \sum_{k} \alpha_{k,\tau}^{j \to i} \cdot \boldsymbol{r}_{k,\tau}^{j \to i} \qquad (13)$$

and update node representation $\boldsymbol{h}_{\tau}^{i}$ as in Eq. (4). We initialize the hub representations at layer $l = 0$ with random trainable parameters. While similar methods to amortize the cost of the attention layer have been explored in different contexts [29, 34], to the best of our knowledge there are no analogous methods tackling efficient computation of spatiotemporal attention coefficients in STGNNs.

## C   Detailed experimental setup

In this appendix, we discuss in detail the experimental settings. We use the same setup of Cini et al. [6][3],[4]. We refer to [6] for details on these baselines.

For SPIN, we use the same hyperparameters in all datasets: $L = 4$ layers; hidden size $d_h = 32$; 2 layers with hidden size 32 for every MLP; ReLU activation functions. Masking out tokens in the target set allows SPIN to propagate only valid information. As a downside, this results in blocking the flow of information on paths through points in the target set. This can be problematic when the input observations are extremely sparse. Nonetheless, it is reasonable to assume that, after only a few propagation steps, the available information has already been partially diffused to locations with missing observations. At this point, blocked paths can be unlocked, allowing for reaching higher-order neighborhoods. In practice, we introduce a hyperparameter $\eta = 3$ to control the number of layers with masked connections and effectively split the propagation process into two phases. It is important to notice that what is being propagated in the second phase are learned representations, not observations (unavailable for masked tokens).

For SPIN-H, we use similar hyperparameters, but 5 layers (with $\eta = 3$); $K = 4$ hubs per node with $d_z = 128$ units each. These hyperparameters have been selected among a small subset of options on the validation set; we expect far better performance to be achievable with further hyperparameter tuning. Depending on the dataset, the number of parameters ranges from $\approx 55K$ to $\approx 95K$ for SPIN and $\approx 540K$ to $\approx 800K$ for SPIN-H. We use Adam optimizer [35], learning rate $lr = 0.0008$ and a cosine scheduler with a warm-up of 12 steps and (partial) restarts every 100 epochs. We train our models with 300 mini-batches of 8 random samples per epoch, fixing the maximum number of epochs to 300 and using early stopping on the validation set with patience of 40 epochs. Due to constraints on memory capacity on some of the GPUs (see the description of the hardware resources below), for SPIN-H we set the batch size to 6 and 16 in AQI and AQI-36, respectively.

To train SPIN-based models, we minimize the following loss function:

$$\mathcal{L} = \sum_{l=1}^{L} \frac{\sum_{\boldsymbol{q}_{\tau}^{i} \in \mathcal{Y}_{t:t+T}} \ell\left(\hat{\boldsymbol{x}}_{\tau}^{i,(l)}, \boldsymbol{x}_{\tau}^{i}\right)}{|\mathcal{Y}_{t:t+T}|}, \qquad (14)$$

where $\ell(\,\cdot\,,\,\cdot\,)$ is the absolute error and $\hat{\boldsymbol{x}}_{\tau}^{i,(l)}$ is $l$-th layer imputation for the $i$-th node at time step $\tau$. Note that, to provide more supervision to the architecture, the loss is computed and backpropagated

---

[3] https://github.com/Graph-Machine-Learning-Group/grin
[4] https://github.com/TorchSpatiotemporal/tsl

w.r.t. representations learned at each layer, not only at the last one. The error is computed only on data not seen by the model at each forward pass. For this reason, we randomly remove $p$ ratio of the input data for each minibatch sample, with $p$ sampled uniformly from $[0.2, 0.5, 0.8]$, and use them to compute the loss. We never use data masked for evaluation to train any model.

For the spatiotemporal Transformer baseline, we use the same training strategy and a similar hyperparameters configuration of SPIN-H: $L = 5$ layers; 4 attention heads; hidden size and feed-forward size of 64 and 128 units, respectively. For SAITS, we use the code provided by the authors[5]. Hyperparameters for SAITS have been selected on the validation set with a random search by using hyperparameter ranges from the original paper.

We recall that the time and memory complexities of SPIN and SPIN-H scale with $O((N + E)T^2)$ and $O((N + E)KT)$, respectively. For the sake of comparison, here we also report the asymptotic complexities of the spatiotemporal Transformer and GRIN. The Transformer alternates temporal attention (i.e., $O(NT^2)$) and spatial attention (i.e., $O(TN^2)$), with a resulting $O((N + T)NT)$ complexity. Let $R$ be the spatial receptive field (i.e., number of graph convolution layers) of the inner MPGRU cell, the time complexity required to process a single direction in GRIN scales with $O(TRE)$. Note also that while most of the operations in the attention-based models can be executed in parallel, GRIN would need to process the entire sequence recurrently, with a consequent performance slowdown at execution time.

All the models were developed in Python [36] using PyTorch [37], PyG [38] and Torch Spatiotemporal [39]. We use Neptune[6] [40] for experiments tracking. All the experiments have been run in a cluster using GPU-enabled nodes with different hardware setups. Running times of SPIN-H training on a node equipped with a 12GB NVIDIA Titan V GPU range from 4 to 14 hours (depending on the dataset). For SPIN we used a node with 40GB NVIDIA A100 GPU, with running times ranging from 4 to 26 hours.

## D   Datasets

In this appendix, we provide details on datasets and preprocessing used for the experiments. We use temporal windows of $T = 24$ steps for all datasets except AQI-36, for which we set $T = 36$. For traffic datasets, we split the data sequentially as 70% for training, 10% for validation, and 20% for testing. For air quality datasets, following Yi et al. [31], we consider as the test set the months of March, June, September, and December and we use valid observation $x_\tau^i$ as ground-truth if the value is missing at the same hour and day in the following month. For data preprocessing we use the same approach of Cini et al. [6], by normalizing data across the feature dimension (graph-wise for graph-based models) to zero mean and unit variance.

In line with [3, 6], we obtain the adjacency matrix from the node pairwise geographical distances using a thresholded Gaussian kernel [41]

$$a^{i,j} = \begin{cases} \exp\left(-\frac{\text{dist}(i,j)^2}{\gamma}\right) & \text{dist}(i, j) \leq \delta \\ 0 & \text{otherwise} \end{cases}, \tag{15}$$

where $\text{dist}(\cdot, \cdot)$ is the geographical distance operator, $\gamma$ is a shape parameter and $\delta$ is the threshold.

Note that we considered settings where the topology is static. The extension to dynamic graphs, where $\boldsymbol{A} = \boldsymbol{A}_t$ and $N = N_t$, can be an interesting future work while being outside the scope of this paper.

## E   Ablation study

Table 3 shows the results of an ablation study on METR-LA (Point missing) and AQI-36. Here, we evaluate the performance in terms of mean absolute error (MAE) and mean relative error (MRE). We consider two different versions of SPIN-H in which we remove the spatiotemporal cross-attention and the temporal self-attention components, respectively. We also report the performance of SPIN, SPIN-H and the Transformer for reference. Results clearly show that both components contribute

---

[5] https://github.com/WenjieDu/SAITS
[6] https://neptune.ai/

**Table 3:** Ablation study to assess the contribution of the single components in the spatiotemporal attention block. Performance averaged over multiple independent runs.

| | METR-LA (P) | | AQI-36 | |
|---|---|---|---|---|
| | MAE | MRE (%) | MAE | MRE (%) |
| **SPIN** | **1.90** $\pm$ **0.01** | **3.29** $\pm$ **0.01** | **11.77** $\pm$ **0.74** | **16.56** $\pm$ **1.05** |
| **SPIN-H** | 1.96 $\pm$ 0.04 | 3.39 $\pm$ 0.06 | **11.08** $\pm$ **0.06** | **15.60** $\pm$ **0.09** |
| Without cross-attention | 2.18 $\pm$ 0.01 | 3.78 $\pm$ 0.01 | 15.36 $\pm$ 0.09 | 21.62 $\pm$ 0.13 |
| Without self-attention | 2.24 $\pm$ 0.09 | 3.88 $\pm$ 0.16 | 13.63 $\pm$ 0.23 | 19.19 $\pm$ 0.32 |
| Transformer | 2.16 $\pm$ 0.00 | 3.74 $\pm$ 0.01 | 11.98 $\pm$ 0.53 | 16.87 $\pm$ 0.75 |

positively to imputation accuracy. We also point out that in METR-LA (P) observations are masked out uniformly at random while the mask in AQI-36 reflects the empirical distribution of missing data in the dataset.

