# OpenReview forum: "Learning to Reconstruct Missing Data from Spatiotemporal Graphs with Sparse Observations"
_logconference.io/LOG/2022/Conference — LoG 2022 Poster_

### Official Review · Reviewer_5BtN · 2022-10-15

**Overall Score:** 8
**Confidence:** 5

**Review:**

Spatio-temporal graphs are usually highly sparse and time series are characterized by multiple, concurrent, or even long sequences of missing data, which motivates this paper to propose a novel class of attention-based architectures that, given a set of highly sparse discrete observations, learn a representation for points in time and space by exploiting a spatiotemporal propagation architecture aligned with the imputation task. This paper argues that biased values inferred through spatial and temporal propagation, coupled with noisy observations, may exacerbate the accumulation of errors in highly sparse data and contribute to hidden state drift similar to the GRIN model.

The so-called Spatiotemporal Point Inference Network (SPIN) learns a parameterized propagation process via combining with Spatiotemporal Cross-attention, Temporal Self-attention, Skip Connection

Points that can be improved:
1. ablation study: reviewer may be curious about the influence of removing the skip connection module. And the ablation study part needs to move to the main content as it can give a straight insight into which component is the most important part in terms of missing values.
 2.  Related work: missing data is a common phenomenon in wild time series data, the reviewer may want to figure out how SPIN can help on other time series tasks, such as forecasting[1,2] tasks.

-------------
1. Cao, Defu, et al. "Spectral temporal graph neural network for multivariate time-series forecasting." Advances in neural information processing systems 33 (2020): 17766-17778.
2.Kim, Jinhee, et al. "End-to-end multi-task learning of missing value imputation and forecasting in time-series data." 2020 25th International Conference on Pattern Recognition (ICPR). IEEE, 2021.

---

### Official Review · Reviewer_fr5V · 2022-10-19

**Overall Score:** 5
**Confidence:** 4

**Review:**

It's a GNN-based model to dynamically impute missing values in multivariate time series. While estimating the missing values, missing patterns (i.e., binary missing mask), cross-sensor relationships, and sensor-level covariates are considered. Overall, this work seems great. Here are several questions for further improvement:

1. Line 46, what do you mean by d-dimensional measurements at time step t? One sensor can have multiple dimensional measurements at a single time step?

2. The sensor level covariates is crucial in your model, however, less explanation is made. For example, the most important questions are:1) how the Q is generated, or for a given multivariate time series, how to get Q? 2) What does Q means, what's each element in it mean?

3. Regarding evaluation, to test your model, you should have a complete sequence to calculate the reconstruction error, not using autoregressive-based proxy.  In contrast, it's highly suggested to measure the RMSE (or other reconstruction error) between the imputed and ground truth time series.

----------------------------------------------------------
Update after reading authors' response:

Thanks for the very brief answers, however, which arouses more concerns. First, I don't understand how can a sensor has multiple dimensional measurements at a single time step. At each time stamp (i.e., a sampling point), a sensor should only provide one readout. As an exemption, for instance, a 3-axis accelerometer is regarded as 3 sensors in data processing. Moreover, the answer to the third question is not convincing. If there's a misunderstanding (which is probably true), the response didn't help me to understand it correctly.

After consideration, I'd like to update my rating to weak reject.

---

### Official Review · Reviewer_ajb9 · 2022-10-21

**Overall Score:** 6
**Confidence:** 4

**Review:**

————————————————————

Summary of the work

This short paper deals with the important problem of reconstructing multivariate time-series on graphs from
so-termed sparse observations (meaning a large fraction of samples are missing). The proposed data imputation
approach is based on a novel attention mechanism that takes data missingness into account by exploiting a
spatiotemporal propagation process to learn predictive representations of the unobserved samples. The method is
dubbed Spatiotemporal Point Inference Network (SPIN). By bringing to bear a hierarchical attention mechanism, it
is shown that space and time complexities can be reduced. Results on several benchmark datasets show the
effectiveness of the proposed imputation technique under different missing data distributions and rates.

————————————————————

Strengths

+ This extended abstract tackles a problem of broad practical importance in time-series analysis. The survey of
related work is appropriate, with a focus on most recent approaches based on deep learning. The GRIN architecture
is claimed to be the state-of-the-art in spatiotemporal imputation, and numerical experiments show that SPIN can
outperform this baselines in highly sparse regimes (i.e., when rate of data missingness is high).
+ While simple and arguably not transformative, an interesting attention layer is proposed to propagate information at
the level of single observations. This method can be interpreted as performing attention in a space-time product graph.
+ Numerical experiments are effective in demonstrating the effectiveness of the proposed approach. The baselines
adopted are reasonable. The three real-world datasets considered appear to be standard in the recent multivariate
time-series literature. The ablation study in Appendix D is particularly revealing on the merits of the spatiotemporal
cross-attention and temporal self-attention modules. Nicely done.

————————————————————

Weaknesses

- The focus of the paper is on sensor network applications. While it is fine to anchor the narrative in a concrete use
case, not exploring other applications seems like a missed opportunity to better convey the impact of the work. The
relevance of this problem is much broader.
- While the paper is for the most part clearly written, Section 3 is so compact that it is at times hard to grasp all technical
details of the proposed model. I found the paragraph on Hierarchical attention particularly hard to follow.
- There are some important yet unjustified claims, for instance that “GRIN suffers from the error propagation typical of autoregressive
models.” At least a reference where that phenomenon has been observed or theoretically established is needed.
The numerical experiments here do not explicitly demonstrate how that behavior is limiting GRIN’s performance. All in all,
such strong statements should be justified, recalibrated, or else removed.
- The problem formulation does not account for dynamic graphs, namely the adjacency matrix is assumed to remain
invariant over the observation horizon. If the graph topology changes, and say the changes are known, can these be
seamlessly accommodated in SPIN? While this could be beyond the scope of this short paper, at least noting
this limitation is important.
- The graph construction approach (using pairwise distances among sensors) is too simplistic and can be problematic
if sensors, say, are at different altitudes (for the pollution dataset for instance). Something as simple as correlation
network may be more appropriate, but the effect of the adopted graph structure on performance is not studied here.
A different experimental test case beyond sensor networks, say where nodes are not tied to points in Euclidean
space would have raised this potential issue of choosing a good graph more prominently.

————————————————————

Recommendation and justification

For the reasons outlined above, the impression is that this paper can be accepted to the extended-abstract track. The
technical innovation is limited but sufficient; on the experimental side results show that the proposed approach can
outperform the state-of-the-art especially in sparse regimes. While computational complexity can be an issue, a lightweight
approach based on hierarchical attention is proposed and shown to be effective (it is unclear whether the idea behind
hierarchical attention is novel, or simply something borrowed from [27]). The paper focuses exclusively on sensor network
applications, and accordingly it feels that an opportunity was missed in better conveying the broader applicability of SPIN.

————————————————————

Questions and suggestions for improvement

- I suggest to spell out the contributions in the Introduction, or after the description of the Related Work.
- Likewise, the paper would benefit from a discussion on limitations of the approach (possibly in the concluding summary).
- Please add a reference to support the claim that GRIN suffers from error accumulation.
- I suggest to elaborate on novelty of the hierarchical attention mechanism relative to the ideas in [27].
- Please define the notation $\mathbf{M}\_{t}$ , $\mathbf{X}\_{<t}$ , $\mathbf{X}\_{>t}$ , and $\mathcal{N}(i)$.
- Is the MLP from (2) somewhere in Figure 1? I can see the MLP from (4), and was wondering where are the $\mathbf{r}$
messages generated.
- The explanation of the hierarchical attention mechanism needs to be revamped. I understand there may be limited space
in the main body, but maybe and additional diagram in the Appendix showing the differences between SPIN and SPIN-H could
help to this end.
- Minor typo in Section 4. Please add a missing space in “previous works[6].”

---

### Official Review · Reviewer_seag · 2022-10-25

**Overall Score:** 1
**Confidence:** 5

**Review:**

The paper focuses on the problem of imputing missing values in a spatio-temporal network.  This is a practical problem, since most real applications are plagued with missing data issues.

Strength:
* The paper provides a good review of recent literature.
* The presentation is modest.
* Presentation experimental results on multiple datasets and compares against several methods.

Weakness:
* The novelty is really limited.  It proposes one more way to learn spatio-temporal dependencies via attention mechanism, and the number of such papers is too numerous to cite.
* The authors try to break down a longer draft into four pages.  It is important to remember that a short paper submission should not be a a sampled version of a longer paper, but needs to deliver a core message and elaborate on it.

A short paper like this could drill into experiments that empirically emphasize the shortcoming of existing methods and provide a detailed analysis, or present a really novel idea with some really exciting results.  It is unlikely that revisiting similar methods on those three standard datasets is going to lead to a big idea.  Does the PEMS-BAY need more complicated methods to break now ground?  The MAE is for the AQI dataset is not that great, but is it a limitation of methods, or a fundamental problem with informational observability?   It may be worthy to investigate how to exploit unique characteristics of different spatio-temporal systems to drive the development of newer methods.  For example, some spatio-temporal networks can exhibit dynamical system properties that make imputation difficult.  Spatio-temporal networks such as wind-energy networks have high temporal variability or relationship with other external signals that makes imputation challenge.  An extended abstract can also be an opportunity to look beyond datasets that are studied extensively and introduce newer datasets.

---

### Meta-Review · Area_Chair_vt4A · 2022-11-21

**Confidence:** 5
**Recommendation:** Reject

**Meta Review:**

This manuscript studies the problem of missing values in multivariate time series and uses spatio-temporal relationships to learn representations and impute missing values.

Key arguments driving discussion among PC members include:

* *Limited novelty and technical innovation.* Three out of four reviewers raised concerns about performing attention in a space-time product graph. Approaching the problem by training a GNN on nodes representing a pair of (nodes, timesteps) and learning spatio-temporal dependencies via attention mechanism has been previously explored in the literature.

* *Further information on related work.* Reviewers would appreciate more information on how the issue of missing data is tackled in current time series literature, including how the new approach relates to the existing literature in the time series field and how it can help with key time series tasks such as forecasting.

* *Experiments could be improved.* Reviewers would appreciate the authors exploring other applications of their approach to better convey the impact of the work. Ablation studies could be improved with additional experiments that would ablate the skip connection module. The consensus was to include another experimental test case beyond sensor networks.

---

### Decision · Program_Chairs · 2022-11-22

Accept (Poster)